# Taxonomic and Phylogenetic Updates on *Apiospora*: Introducing Four New Species from *Wurfbainia villosa* and Grasses in China

**DOI:** 10.3390/jof9111087

**Published:** 2023-11-06

**Authors:** Chunfang Liao, Indunil Chinthani Senanayake, Wei Dong, Kandawatte Wedaralalage Thilini Chethana, Khanobporn Tangtrakulwanich, Yunxia Zhang, Mingkwan Doilom

**Affiliations:** 1Innovative Institute for Plant Health, Key Laboratory of Green Prevention and Control on Fruits and Vegetables in South China, Ministry of Agriculture and Rural Affairs, Zhongkai University of Agriculture and Engineering, Guangzhou 510225, China; 6371105002@lamduan.mfu.ac.th (C.L.); indunilchinthani@gmail.com (I.C.S.); dongwei0312@hotmail.com (W.D.); kandawatte.thi@mfu.ac.th (K.W.T.C.); yx_zhang08@163.com (Y.Z.); 2Center of Excellence in Fungal Research, Mae Fah Luang University, Chiang Rai 57100, Thailand; 3School of Science, Mae Fah Luang University, Chiang Rai 57100, Thailand; khanobporn.tan@mfu.ac.th

**Keywords:** Asia, Amphisphaeriales, Apiosporaceae, endophytes, saprobes, taxonomy

## Abstract

*Apiospora*, an ascomycetous genus in Apiosporaceae, comprises saprobes, endophytes, and pathogens of humans and plants. They have a cosmopolitan distribution with a wide range of hosts reported from Asia. In the present study, we collected and isolated *Apiospora* species from *Wurfbainia villosa* and grasses in Guangdong and Yunnan provinces in China. Multi-locus phylogeny based on the internal transcribed spacer, the large subunit nuclear rDNA, the partial translation elongation factor 1-*α*, and β-tubulin was performed to clarify the phylogenetic affinities of the *Apiospora* species. Based on the distinctive morphological characteristics and molecular evidence, *Ap*. *endophytica*, *Ap*. *guangdongensis*, *Ap*. *wurfbainiae*, and *Ap*. *yunnanensis* are proposed. Descriptions, illustrations, and notes for the newly discovered species are provided and compared with closely related *Apiospora* species. An updated phylogeny of *Apiospora* is presented, along with a discussion on the phylogenetic affinities of ambiguous taxa.

## 1. Introduction

Recent advances in fungal taxonomy and phylogeny have resulted in taxonomic revisions in numerous genera [1,2,3,4], including *Apiospora*. *Apiospora* belongs to Apiosporaceae, Amphisphaeriales, Sordariomycetes, and Ascomycota [5]. It was introduced by Saccardo [6], but the typification was not indicated. Subsequently, Clements and Shear [7] designated *Apiospora montagnei* Sacc. as the type species. However, Crous and Groenewald [8] synonymized *Ap. montagnei* under *Arthrinium arundinis* based on the presence of similar characters in their sexual morphs, including multi-locular perithecial stromata and hyaline ascospores surrounded by a thick gelatinous sheath, and also considering that *Arthrinium* is an older and more commonly referred to name than *Apiospora* [8,9,10,11]. Crous and Groenewald [8], therefore, treated the sexual genus *Apiospora* as a synonym of *Arthinium* on the basis that *Arthinium* is earlier proposal and in more frequent usage [10,11]. This taxonomic treatment has been followed by several studies [12,13,14]. Subsequently, Pintos and Alvarado [15] re-evaluated the phylogenetic placements of *Apiospora* and *Arthrinium* based on multi-locus phylogeny using the internal transcribed spacer (ITS), large subunit nuclear rDNA (LSU), the partial translation elongation factor 1-*α* (*tef1-α*), and β-tubulin (*tub2*) sequence data. The result showed that several *Arthrinium* species, including the type species *Ar*. *caricicola*, form a well-supported but distant clade compared to other *Arthrinium* species, indicating them into two independent genera. Therefore, the species within this clade were retained in *Arthrinium*, while other species were transferred to *Apiospora* [15]. *Apiospora* is accepted with conidia that are globose to subglobose in the face view and lenticular in the side view with a pale equatorial slit, whereas *Arthrinium* possesses conidia of various shapes (angular, curved, fusiform, globose, polygonal, and navicular) [15]. The sexual morphs of *Apiospora* are characterized by immersed, dark brown to black, lenticular, or dome-shaped ascostromata that are erumpent through a longitudinal split, unitunicate, broadly clavate to cylindric-clavate asci, and hyaline ascospores that are 1-septate near the lower end, with or without a sheath [13]. Based on the recent taxonomic treatment and multi-locus phylogenetic analyses, sixty-eight species of *Arthrinium* were synonymized under *Apiospora* [14,15,16]. Up to now, 133 epithets are listed under *Apiospora* in the Index Fungorum [17].

Species of *Apiospora* are distributed worldwide, mostly from terrestrial and aquatic habitats in Asia [14,17,18]. They are reported as important plant pathogens causing significant damage to economic plants. For example, *Apiospora arundinis* (previously known as *Arthrinium arundinis*) is a causal agent of leaf edge spot disease of peach (*Prunus persica*) in China, with a 20 to 40% disease incidence in two hectares of a severely infected peach orchard [19]. *Apiospora arundinis* has been commonly reported as the pathogen of *Phyllostachys praecox*, causing brown culm streak [20]. *Apiospora sacchari* is reported to cause Barley kernel blight [21], while *Ap*. *phaeospermum* is a pathogen causing damping-off disease in wheat [22]. In addition, *Apiospora arundinis* and *A. montagnei* have been reported as animal and human pathogens that cause onychomycosis [23,24]. They are also isolated from air and soil, while some are lichen-associated [12,17]. Many *Apiospora* species are known as saprobes and endophytes on many host plants, including thorny bamboo (*Bambusa bambos*), bristlegrass (*Setaria viridis*), loquat (*Eriobotrya japonica*), windmill palm (*Trachycarpus fortunei*), and tea (*Camellia sinensis*) [12,15,16,24,25,26,27,28,29].

In a survey for fungi associated with monocotyledon plants in China, we collected and isolated *Apiospora* strains from *Wurfbainia villosa* and grasses in Guangdong and Yunnan provinces. The identifications of *Apiospora* strains in this study were performed through the combination of ITS, LSU, *tef1-α*, and *tub2* sequence analyses, along with morphological characteristics. A pairwise homoplasy index test was conducted to determine the recombination level within phylogenetically closely related species. The novel *Apiospora* species were identified, following the guidelines in Jeewon and Hyde [30], Maharachchikumbura et al. [31], and Pem et al. [4].

## 2. Materials and Methods

### 2.1. Sample Collection, Observation, and Isolation

Saprobic fungi were collected from dead stems of grasses at the Kunming Institute of Botany, Kunming City, Yunnan Province, China. The samples were placed into zip-lock bags and returned to the laboratory for fungal observation and isolation. The specimens were observed after 2–3 days of inoculation at room temperature using SZ650 (Chongqing Auto Optical Instrument Co., Ltd., Chongqing, China) stereo microscope. Fungal structures (e.g., ascomata, hamathecium, asci, and ascospores) were examined using Nikon Eclipse 80i, connected to the industrial Digital Sight DS-Fi1 (Panasonic, Tokyo, Japan) microscope imaging system. Single spore isolation was performed as described by Senanayake et al. [28]. The germinated spores were grown on potato dextrose agar (PDA: potato 200 g/L, dextrose 15 g/L, agar 15 g/L) and incubated at 25 ± 2 °C for two weeks.

Endophytic fungi were isolated from the healthy leaves of *Wurfbainia villosa* in Yongning town, Yangjiang City, Guangdong Province, China. The isolation procedures of plant materials were performed as described by Senanayake et al. [28]. Briefly, fresh, healthy leaves were gently rinsed with tap water to eliminate any accumulated particulate matter. The leaves were surface sterilized in 2.5% sodium hypochlorite for 1 min, followed by 75% ethanol for 2 min. The samples were subsequently rinsed three times with sterile water for 3 min each time and air-dried using sterile tissue filter paper. The sterilized leaves were then cut into 0.5 × 0.5 cm pieces using sterile scissors and aseptically transferred onto PDA and incubated at 25 °C [28]. The hyphal tips grown from sterilized leaves after three days of incubation were transferred to fresh PDA for three to four times for purification to obtain a pure culture.

All fungal isolates were preserved on PDA slants and stored at 4 °C and in 15% glycerol. The fungal structures were measured using Tarosoft (R) Image Frame Work program v. 0.9.7. and NIS-Elements BR 5.30.03. The living cultures were deposited in the Zhongkai University of Agriculture and Engineering Culture Collection (ZHKUCC), Guangdong, China. Herbarium specimens were deposited in the Mycological Herbarium of Zhongkai University of Agriculture and Engineering (MHZU), Guangzhou, China. The new species were registered in Faces of Fungi (FoF) (http://www.facesoffungi.org; accessed on 17 October 2023) [32] and Index Fungorum (IF) databases (http://www.indexfungorum.org/names/names.asp; accessed on 17 October 2023). The records of Greater Mekong Subregion fungi will be placed in the GMS database [33].

### 2.2. DNA Extraction, PCR Amplification, and Sequencing

Fungal mycelia grown on PDA for 5–7 days were collected for Genomic DNA extraction using the MagPure Plant DNA AS Kit, following the manufacturer’s instructions (Guangzhou Magen Biotechnology Co., Ltd., Guangzhou, China). Extracted DNA was stored at −20 °C. The internal transcribed spacer (ITS), large subunit rDNA (LSU), β-tubulin (*tub2*), and partial translation elongation factor 1–*α* (*tef1-α*) were amplified and sequenced using primer ITS1 and ITS4 [34,35], LR5 and LR0R [36], BT2a and BT2b [37], and EF1-728F and EF2 [38,39], respectively.

The 25 µL volume of Polymerase chain reaction (PCR) contains 12.5 µL 2 × Taq Master Mix (buffer, dNTPs, and Taq; Nanjing Vazyme Biotech Co., Ltd., Nanjing, China), 9.5 µL of ddH2O, 1 µL of each primer, and 1 µL of DNA template. The PCR thermal cycle program for ITS and LSU amplification was conducted with an initial denaturation at 95 °C for 3 min, followed by 35 cycles of 94 °C for 30 s; the annealing temperature was 52 °C for 30 s for ITS and LSU; 72 °C for 1 min; and final elongation at 72 °C for 10 min. The annealing temperatures were adjusted to 53.5 °C (30 s) and 55 °C (45 s) for *tub2* and *tef1-α*, respectively. PCR products were purified and sequenced by Tianyi Huiyuan Gene Technology & Services Co. (Guangzhou, China). All sequences generated in this study were submitted to GenBank [40].

### 2.3. Phylogenetic Analyses

The sequence quality of obtained sequences was assured by checking chromatograms using Bioeidit v. 7.2.3 [41]. Sequences used for phylogenetic analysis were downloaded from GenBank according to the Blastn search of ITS in the GenBank database and following the published literature [16]. A total of 191 sequences were used in the phylogenetic analysis (Table 1). *Sporocadus trimorphus* strains CFCC 55171 and ROC 113 were used as outgroup taxa. Four loci, ITS, LSU, *tef1-α*, and *tub2*, were aligned in MAFFT version v. 7 online program [42] and edited manually where necessary using BioEdit v. 7.2.3 [41]. Alignments were converted to NEXUS format using Alignment Transformation Environment online platform (http://www.sing-group.org/ALTER/; accessed on 17 October 2023).

Maximum likelihood (ML) and Bayesian inference (BI) analyses were performed in the CIPRES Science Gateway online platform [43] based on the combined ITS, LSU, *tef1-α*, and *tub2* sequence data. The ML analysis was carried out with GTR+G+I evolutionary substitution using RAxML-HPC v.8.2.12 on XSEDE (https://www.phylo.org/; accessed on 17 October 2023) [44], with 1000 rapid bootstrap inferences, followed by a thorough ML search. All free model parameters were estimated by RAxML ML of 25 per site rate categories. The likelihood of the final tree was evaluated and optimized under GAMMA. Bayesian Inference (BI) analysis was conducted using the Markov Chain Monte Carlo (MCMC) method and performed in MrBayes XSEDE (3.2.7a) [45]. Six simultaneous Markov chains were run for 2,000,000 generations, and the trees were sampled for each 100th generation. Phylogenetic trees were visualized in FigTree v. 1.4.0 [46] and formatted using PowerPoint 2010 (Microsoft Corporation, WA, USA).

### 2.4. Pairwise Homoplasy Index (PHI)

A pairwise homoplasy index (PHI) test [47] was performed using SplitsTree v. 4.15.1 [48] to determine the recombination level within phylogenetically closely related species of the new strains in this study (*Apiospora endophytica*, *A. guangdongensis*) with *A. arundinis*, *A. aurea*, *A. cordylies*, and *A. hydei*. The combined ITS, LSU, *tef1-α*, and *tub2* of these phylogenetically closely related species were applied for PHI test and analyses. The PHI results (Φw) > 0.05 indicated no significant recombination in the dataset. The relationships between our strains with closely related taxa were visualized by constructing splits graphs using Log-Det transformation and split decomposition options using SplitsTree v. 4.15.1.

## 3. Results

### 3.1. Phylogeny

The phylogenetic tree was constructed based on the combined ITS, LSU, *tef1-α*, and *tub2* sequence data of 191 strains (including our new strains), with *Sporocadus trimorphus* strains CFCC 55171 and ROC 113 as outgroup taxa. There are a total of 2936 characters, including gaps (ITS: 1–772, LSU: 773–1621, *tef1-α*: 1622–2309, *tub2*: 2310–2936). The topology of the ML analysis was similar to the BI analysis, and the best-scoring RAxML tree with a final ML optimization likelihood value of -36321.892470 is presented (Figure 1). The matrix had 1805 distinct alignment patterns, with 38.08% undetermined characters or gaps. Estimated base frequencies were as follows: A = 0.208057, C = 0.296775, G = 0.242495, T = 0.252673; substitution rates AC = 1.090339, AG = 3.411914, AT = 1.286700, CG = 0.887072, CT = 4.062650, GT = 1.000000; gamma distribution shape parameter α = 0.777262. Phylogenetic analyses showed that our strains belong to *Apiospora*. The isolates ZHKUCC 23-0010 and ZHKUCC 23-0011 had a close affinity to *Apiospora phyllostachydis* (MFLUCC 18-1101) with 100% ML bootstrap support and 1.00 BYPP. The isolates ZHKUCC 23-0004 and ZHKUCC 23-0005 formed a sister to *A. arundinis* (CBS 449.92 and CBS 133509) with 100% ML bootstrap support and 1.00 BYPP. Two isolates of ZHKUCC 23-0014 and ZHKUCC 23-0015 formed a distinct lineage and sister to *A. qinlingensis* (CFCC 52303 and CFCC 52304) and *A. koreana* (KUC21332 and KUC21348) with 96% ML bootstrap support and 0.90 posterior probability in BI analysis. The isolates ZHKUCC 23-0012 and ZHKUCC 23-0013 clustered with *A. guizhounese* (LC 5318 and LC 5322) with low support in ML and BI analyses (44% ML and 0.72 BYPP). The isolates ZHKUCC 23-0006 and ZHKUCC 23-0007 formed a sister to *A. hydei* (CBS 114990 and KUMCC 16-0204) with 96% ML bootstrap support and 1.00 BYPP. Two isolates, ZHKUCC 23-0008 and ZHKUCC 23-0009, formed a distinct lineage and sister to *Apiospora* species with 80% ML and 1.00 BYPP (Figure 1).

### 3.2. A Pairwise Homoplasy Index

The recombination level within phylogenetically closely related species of generated strains of *Apiospora endophytica* with *A. aurea*, *A. cordylines*, and *A. hydei* as well as phylogenetically closely related species of *A. guangdongensis* with *A. arundinis* were implied in a pairwise homoplasy index (PHI) test using combined ITS, LSU, *tef1-α*, and *tub2* sequence dataset. The PHI result showed that there was no evidence of significant recombination (Φw = 0.06901) among *A*. *endophytica*, *A. aurea*, *A. cordylines*, and *A. hydei* with the combined dataset (Figure 2). The *A*. *guangdongensis* and *A*. *arundinis* has also no significant evidence of recombination (Φw = 1.00) (Figure 3).

### 3.3. Taxonomy

***Apiospora endophytica*** C.F. Liao and Doilom, sp. nov. Figure 4.

Index Fungorum number: IF900356; Facesoffungi number: FoF14658.

Etymology: The epithet “*endophytica*” refers to the endophytic lifestyle of the species.

*Endophytic* in leaves of *Wurfbainia villosa*. Sexual morph: undetermined. Asexual morph: sporulating on PDA after one month, spore mass visible as black, scattered on white colonies. *Hyphae* 2–5 μm wide ($\bar{X}$ = 2.5 μm, n = 30), branched, hyaline to golden brown, septate, smooth-walled. *Conidiophores* reduced to conidiogenous cells. *Conidiogenous cells* 4–14 × 2–7 μm ($\bar{X}$ = 7.5 × 5 μm, n = 35), aggregated in clusters or solitary, hyaline to golden brown, erect, unbranched, cylindrical or clavate, ampulliform or obtriangular, and smooth-walled. *Conidia* 14–19 × 12–18 μm ($\bar{X}$ = 17 × 15 μm, n = 30) in the face view, 11–19 × 9–16 μm ($\bar{X}$ = 15 × 12 μm, n = 20) in the side view, initially hyaline, becoming pale brown to dark brown, globose to subglobose, obovoid to ellipsoidal in the face view, lenticular with a thick equatorial slit in the side view, and smooth-walled. *Sterile cells* not observed.

Culture characteristics: colonies on PDA reached 2.6 cm in one week at 28 ± 2 °C, fluffy, spreading, with dense, aerial mycelium, composed of small bumps, forming a circle around the center, surface and reverse both golden yellow in the center, and turning white at the edge.

Material examined: China, Guangdong Province, Yangjiang City, Yongning town, 24°40′53″ N 118°41′31″ E, asymptomatic leaves of *Wurfbainia villosa* (Lour.) Škorničk. and A.D. Poulsen (Zingiberaceae), 1 October 2021, Chunfang Liao, (ZHKU 23-0002, holotype, dried culture); ex-type living culture ZHKUCC 23-0006, ibid., and living culture ZHKUCC 23-0007.

Notes: In the phylogenetic analyses (Figure 1), *Ap. endophytica* (ZHKUCC 23-0006, ZHKUCC 23-0007) clustered sister to *Ap. hydei* (CBS 114990 and KUMCC 16-0204) with 96% ML bootstrap support and 1.00 BYPP and formed a distinct lineage separated from *Ap. cordylines* (GUCC 10026) with 100% ML bootstrap support and 1.00 BYPP) and *Ap. aurea* (CBS 244.83) by 100% ML bootstrap support and 1.00 BYPP. Morphologically, conidiogenous cells of *Ap. endophytica* are cylindrical or clavate, ampulliform or obtriangular, while they are subcylindrical to doliiform to lageniform in *Ap. hydei*. The conidia of *Ap. endophytica* are dark brown and smooth, while they are brown and roughened in *Ap. hydei*. In addition, *Ap. endophytica* has larger conidiogenous cells compared to than those of *Ap. hydei* (4–14 × 2–7 μm vs. 5–8 × 4–5 μm). *Apiospora endophytica* differs from *Ap. cordylines* and *Ap. aurea* based on the size and shape of conidiogenous cells and conidia (Table 2). The PHI test results indicated no significant recombination between *Ap. endophytica* and closely related species *Ap. aurea* (CBS 244.83), *Ap*. *cordylies* (GUCC 10026), and *Ap. hydei* (CBS 114990) (Figure 2). Both morphological and molecular evidence supported *Ap. endophytica* as a new species.

***Apiospora guangdongensis*** C.F. Liao and Doilom, sp. nov. Figure 5.

Index Fungorum number: IF900357; Facesoffungi number: FoF14659.

Etymology: The epithet “*guangdongensis*” refers to the locality, Guangdong Province, China where the holotype was collected.

*Endophytic* in asymptomatic leaves of *Wurfbainia villosa*. Sexual morph: undetermined. Asexual morph: sporulated on PDA after one month, spore mass visible as black, scattered to aggregated on white colonies. *Hyphae* 2–3 μm diam. ($\bar{X}$ = 2.5 μm, n = 30), branched, hyaline, septate, smooth, thin-walled, forming hyphal coils. *Conidiophores* 45–53 × 2–4 μm ($\bar{X}$ = 49 × 2.5 μm, n = 30), micronematous, mononematous, erect, solitary, subcylindrical, unbranched, straight or flexuous, hyaline, smooth-walled, sometimes reduced to conidiogenous cells. *Conidiogenous cells* 4–9 × 2–5 μm ($\bar{X}$ = 6 × 3.5 μm, n = 30), arising from hyphae, aggregated in clusters or solitary, terminal or lateral, smooth, straight or slightly curved, cylindrical or ampulliform, and sometimes ovate or obpyriform. *Conidia* 6–9 × 5–9 μm ($\bar{X}$ = 8 × 7 μm, n = 30) in the face view, 5–8 × 4–6 μm ($\bar{X}$ = 6.5 × 5 μm, n = 30) in the side view, initially hyaline, becoming pale brown to dark brown, globose to ellipsoidal in face view, lenticular with broad equatorial slit in the side view, aseptate, smooth-walled. *Sterile cells* 9–16 × 3–8 μm ($\bar{X}$ = 12 × 5 μm, n = 30), light brown, elongate. *Chlamydospores* produced in chain, terminal, globose to subglobose, hyaline, smooth-walled.

Culture characteristics: colonies on PDA reaching 6.6 cm in one week at 28 ± 2 °C, floccose, sparse, concentrically spreading, forming aerial mycelia, edge irregular, surface pale brown in center, white at the edge, with punctate or flaky black spores, reverse white to pale brown with some pale brown spot, no pigment.

Material examined: China, Guangdong Province, Yangjiang City, Yongning town, 24°40′53″ N 118°41′31″ E, asymptomatic leaves of *Wurfbainia villosa* (Lour.) Škorničk. and A.D. Poulsen (Zingiberaceae), 1 October 2021, Chunfang Liao, (ZHKU 23-0001, holotype, dried culture); ex-type cultures ZHKUCC 23-0004, ibid., living culture ZHKUCC 23-0005.

Notes: The phylogenetic analyses showed that *Ap. guangdongensis* (ZHKUCC 23-0004 and ZHKUCC 23-0005) formed a sister branch to *Ap. arundinis* with 100% ML bootstrap support and 1.00 BYPP (Figure 1). The morphology of *Ap. guangdongensis* differs from *Ap. arundinis* by having shorter conidiogenous cells (4–9 × 2–5 μm vs. 6–12 × 3–4 μm) and larger conidia (6–9 × 5–9 μm vs. (5–)6–7 μm in the face view, 5–8 × 4–6 μm vs. 3–4 μm) [8]. The conidiogenous cells of *Ap. guangdongensis* are cylindrical or ampulliform, sometimes ovate or obpyriform, while they are ampulliform in *Ap. arundinis*. The result of the PHI test showed no significant recombination between our isolates and *Ap*. *arundinis* (Figure 3). Based on distinct morphological and molecular evidence, we propose *Ap. guangdongensis* as a new species.

***Apiospora wurfbainiae*** C.F. Liao and Doilom, sp. nov. Figure 6.

Index Fungorum number: IF900355; Facesoffungi number: FoF14660.

Etymology: The epithet “*Wurfbainiae*” refers to the host genus *Wurfbainia*, from which the holotype was collected.

*Endophytic* in asymptomatic leaves of *Wurfbainia villosa*. Sexual morph: undetermined. Asexual morph: sporulated on PDA after three months, spore mass visible as black, scattered on colonies. *Hyphae* 1–3 μm diam. ($\bar{X}$ = 2 μm, n = 30), branched, hyaline, septate, smooth, forming hyphal coils. *Conidiophores* reduced to conidiogenous cells, hyaline, smooth, branched. *Conidiogenous cells* 7–50 × 2–8 μm ($\bar{X}$ = 22 × 5 μm, n = 60), holoblastic, monoblastic, discrete, hyaline, straight or curved, cylindrical to lageniform, smooth-walled. *Conidia* 7–9 × 5–9 μm ($\bar{X}$ = 8 × 7 μm, n = 30) in the face view, 6–9 × 3–6 μm ($\bar{X}$ = 7 × 4.5 μm, n = 20) in the side view, obovoid, globose to subglobose in face view, lenticular with pale equatorial slit in the side view, initially hyaline, becoming pale brown to dark brown, multi-guttulate, smooth-walled. *Sterile cells* 8–31 × 2–12 μm ($\bar{X}$ = 14 × 5 μm, n = 30), light brown, elongated, cylindrical, ovate, triangular-shaped.

Culture characteristics: colonies on PDA reaching 6.8 cm in one week at 28 ± 2 °C, flatted, dense mycelium, edge regular, gray in the center, with some white globular spots from above; pale yellow to gray with some orange spots from below.

Material examined: China, Guangdong Province, Yangjiang City, Yongning town, 24°40′53″ N 118°41′31″ E, asymptomatic leaves of *Wurfbainia villosa* (Lour.) Škorničk. and A.D. Poulsen (Zingiberaceae), 1 October 2021, Chunfang Liao, (ZHKU 23-0003, holotype, dried culture); ex-type living culture ZHKUCC 23-0008, ibid., living culture ZHKUCC 23-0009.

Notes: *Apiospora wurfbainiae* shares morphological similarities to *Ap. guangdongensis* in having globose conidia as well as overlapping conidial size (7–9 × 5–9 μm vs. 6–9 × 5–9 μm in the face view). However, *Ap. wurfbainiae* has larger conidiogenous cells (7–50 × 2–8 μm vs. 4–9 × 2–5 μm) than *Ap. guangdongensis*. The sterile cells of *Ap. wurfbainiae* are elongated, cylindrical, ovate, triangular-shaped while only elongated cells were observed in *Ap. guangdongensis*.

In the phylogenetic analysis (Figure 1), *Ap. wurfbainiae* (ZHKUCC 23-0008, ZHKUCC 23-0009) form a distinct subclade which is basal to *Apiospora* clade with 80% ML and 1.00% BYPP. Further, this subclade is closely related to another subclade consisting of *Ap. tropica*, *Ap. subglobosa*, and *Ap. neosubglobosa*. Morphologically, *Ap. tropica*, *Ap. subglobosa*, and *Ap. neosubglobosa* were described based on their sexual morph but *Ap. wurfbainiae* was identified solely by its asexual morph, thus their morphological characteristics could not be compared. However, molecular evidence clearly separates *Ap. wurfbainiae* from other known *Apiospora* species. Hence, we introduce *Ap. wurfbainiae* as a novel species.

***Apiospora yunnanensis*** C.F. Liao and Doilom, sp. nov. Figure 7.

Index Fungorum number: IF900358; Facesoffungi number: FoF14661.

Etymology: The epithet “*yunnanensis*” refers to the location, Yunnan Province, China where the holotype was collected.

*Saprobic* on dead stem of grass. Sexual morph: *Ascostromata* 750–3600 × 230–420 μm

($\bar{X}$ = 1590 × 290 μm, n = 20), solitary to gregarious, scattered, immersed to erumpent, with the long axis broken at the top, black, ostiolate. *Ascomata* 75–155 × 125–245 μm ($\bar{X}$ = 125 × 200 μm, n = 20), perithecial, immersed, pale brown to black, ampulliform to subglobose with a flattened base in cross-section, 1–2-loculate. *Ostiole* 35–80 μm wide ($\bar{X}$ = 54 μm, n = 20), periphysate, central. *Peridium* 8–26 μm wide (($\bar{X}$= 17 μm, n = 50), 2–5-layered, outer layer composed of brown to dark brown, intermixed with host tissue, thick-walled, inner layer composed of hyaline, thin-walled cells of *textura angularis*. *Hamathecium* 5–13 μm wide ($\bar{X}$ = 9 μm, n = 25), composed of hyaline, septate, unbranched paraphyses, embedded in a gelatinous matrix. *Asci* 70–93 × 15–23 μm ($\bar{X}$ = 81 × 18 μm, n = 30), 8-spored, unitunicate, broadly cylindrical to clavate, apically rounded, with a pedicel. *Ascospores* 21–30 × 6–10 μm ($\bar{X}$ = 23 × 8 μm, n = 50), overlapping 1–2-seriate, clavate to fusiform, 1-septate, composed of a large upper cell and small lower cell, straight to slightly curved near the lower cell, guttulate, hyaline, smooth-walled, and surrounded by a gelatinous sheath. Asexual morph: undetermined.

Culture characteristics: Colonies on PDA reaching 6.0 cm in one week at 28 ± 2 °C, cottony in the center, dense, flat, edge mycelium spars, surface white in center, reverse white to pale brown.

Material examined: China, Yunnan Province, Kunming Institute of botanical garden, 25°02′11″ N 102°42′31″ E, dead stem of grass (Poaceae), 20 July 2019, Chunfang Liao, (ZHKU 23-0004, holotype, dried culture); ex-type living culture ZHKUCC 23-00014, ibid., living culture ZHKUCC 23-00015.

Notes: In the phylogenetic analysis, *Ap. yunnanensis* (ZHKUCC 23-00014, ZHKUCC 23-00015) formed a distinct branch with *Ap. koreana* and *Ap. qinlingensis* with ML = 96%, and BYPP = 0.90% (Figure 1). In comparison between ITS, *tef1-α*, and *tub2* sequence data between our isolate (ZHKUCC 23-00014; ex-type) and *Ap. koreana* (KUC21332; ex-type), there were differences in 9.44% (51/540 bp), 6.85% (32/467 bp), and 9.31% (38/408 bp), respectively, while the comparison with *Ap. qinlingensis* (CFCC 52303; ex-type) showed differences in 13.61% (78/573 bp), 21.9% (97/442 bp), and 10.3% (52/505 bp), respectively. The LSU sequence data are currently unavailable for *Ap. koreana* and *Ap. qinlingensis*. The morphological characteristics of *Ap. yunnanensis* cannot be compared with those of its phylogenetically closely related species, as *Ap. koreana* and *Ap. qinlingensis* were described based on their asexual morph. While *Ap. yunnanensis* is currently known only from its sexual morph, attempts to sporulate its conidia on media with pine needles have been unsuccessful.

Morphologically, *Ap. yunnanensis* is similar to *Ap. montagnei* in having immersed to erumpent ascostromata, with the long axis broken at the top, broadly cylindrical to clavate asci and clavate to fusiform ascospores. However, *Ap. yunnanensis* is distinguished from *Ap. montagnei* by its shorter and wider asci (70–93 × 15–23 μm vs. 72–115 × 14–18 µm) and larger ascospores (20–30 × 6–10 μm vs. 21–25 × 6–8 µm) [15]. The comparison of LSU sequence data from our isolate *Ap. yunnanensis* (ZHKUCC 23-00014) with the sequences identified as *Ap. montagnei* ICMP 6967 and AFTOL-ID 951 in NCBI databases revealed differences of 2.24% (18/804 bp) and 2.28% (18/788 bp), respectively. We hereby propose *Ap. yunnanensis* as a novel species.

## 4. Discussion

The species diversity of *Apiospora* has been $\bar{X}$ expanding steadily, especially in China. To date, 40 *Apiospora* species have been introduced in China, including four novel species in this study [14,16,28,29,51] (Table 1). These four new species, *Ap. endophytica*, *Ap. guangdongensis*, *Ap. wurfbainiae*, and *Ap. yunnanensis*, are introduced based on morphological characteristics and multi-locus phylogenetic analyses. Based on the host diversity of *Apiospora* species reported by Monkai et al. [52], it was found that most *Apiospora* species are associated with Poaceae (63%), including bamboo (31%), non-bamboo (32%), and other plant families (27%). Our study reveals another *Apiospora* species, *Ap. yunnanensis*, which was isolated from grass (Poaceae). Furthermore, the additional three species, *Ap. endophytica*, *Ap. guangdongensis*, and *Ap. wurfbainiae,* have been found on *W. villosa* belonging to the plant family Zingiberaceae. It is likely that *W. villosa* harbors high *Apiospora* species diversity. In addition, several *Apiospora* species have been reported from various monocotyledon plants, including bamboos, *Cordyline fruticose*, grasses, and *Phragmites australis* [8,13,49] (this study). It suggested that monocotyledon plants may harbor a high species diversity of *Apiospora* species.

Our study presents an updated phylogeny for *Apiospora* species, which is the additional contribution of this study to the previous works. By integrating the recent literature from Pintos et al. [8], Tian et al. [16], and Phukhamsakda et al. [53] with our new collections, we recognize 93 species including four newly discovered species based on multi-locus phylogenetic analyses and morphology. However, the phylogenetic analyses of combined ITS, LSU, *tef1-α*, and *tub2* revealed a close phylogenetic relationship between *Ap. hispanica* and *Ap. mediterranea* (Figure 1), which is consistent with the previous studies in Tian et al. [16], Monkai et al. [52], and Phukhamsakda et al. [53]. The comparison of LSU, ITS, and *tub2* sequence data showed that *Ap. hispanica* is identical to *Ap. Mediterranea*; however, their *tef1-α* sequence data are currently unavailable in GenBank. Morphologically, *Ap. hispanica* is similar to *Ap. mediterranea* by having basauxic, macronematous, and mononematous conidiophores, but it has smaller conidia than *Ap. mediterranea* (7.5–8.5 × 6.2–7.6 μm vs. 9–9.5 × 7.5–9 μm) [54]. Our phylogenetic result supports the suggestion of Monkai et al. [52] that the morphological reexamination of the type specimens of *Ap. hispanica* and *Ap. mediterranea,* including their molecular data from additional genes such as *tef1-α,* should be investigated to confirm a putative synonymy.

In addition, *Ap. marina* shares a close phylogenetic affinity with *Ap. paraphaeosperma* and *Ap. rasikravindrae,* and these three species clustered sister to *Ap. acutiapica* and *Ap. pseudorasikravindrae* with 100% ML and 1.00 BYPP support (Figure 1), which is consistent with the phylogenetic result in Monkai et al. [52]. Morphologically, *Ap. marina* is similar to *Ap. paraphaeosperma* and *Ap. rasikravindrae* by having brown, smooth, globose to elongate conidia, but *Ap. marina* has smaller conidia than *Ap. paraphaeosperma* (9.5–)10–12 (−13) × (7.5–)8.0–10 μm vs. 10–19 μm diam.), and *Ap. rasikravindrae* (9.5–)10–12 (−13) × (7.5–)8.0–10 vs. 10−15 × 6.0−10.5 μm) (Appendix A). Regarding the aforementioned factors, we suggest that the species boundaries of these ambiguous species should be re-evaluated to confirm the taxonomic status and to facilitate the identification of species grouped in this clade, and that *tef1-α* and *tub2* sequence data from the ex-type of *Ap. rasikravindrae* (NFCCI 2144) are required. Additionally, there are 41 morphospecies (species without molecular data) listed under *Apiospora* (Appendix A). Pintos and Alvarado [15] examined the lectotype for *Sphaeria apiospora* (=*Ap. montagnei*, type species of *Apiospora*) specimens preserved at the PC fungarium, which were collected from Poaceae in lowland Mediterranean habitats. The taxonomic status of the remaining taxa, lacking sequence information and comprehensive morphological descriptions, remains uncertain and requires further investigation.

In this study, we compiled the available information on the sexual/ asexual morph of *Apiospora* species, including their known lifestyle from the relevant literature (Table 1). According to these data, 12 species have only been reported in their sexual morphs, while 63 species are known solely by their asexual morphs. Additionally, 19 species have been described in both sexual and asexual morphs. The prevalence of *Apiospora* species is likely to be associated with their asexual morph occurring as saprobic and endophytic lifestyles. On the other hand, the sexual morph is commonly observed from saprobic isolates thus far. Moreover, some *Apiospora* species have been reported in several lifestyles. For example, *Ap. arundinis*, *Ap*. *hydei*, *Ap. thailandica*, and *Ap. yunnana* have been reported in both saprobes and endophytes [8,25,55]. In addition, *Ap. arundinis* has been known as a saprobe, endophyte and pathogen [56]. The investigation into the potential transition of endophytic or saprobic of *Apiospora* to alternative lifestyles, such as becoming pathogens, is crucial for understanding their ecological role.

In view of the biological applications, many species of *Apiospora* produce an interesting bioactive secondary metabolite which could be a promising source of pharmacological and medicinal applications. For instance, a saprobic isolate of *Ap. chromolaenae* showed antimicrobial activity against *Escherichia coli* [57]. *Apiospora saccharicola* and *Ap. sacchari* isolated from *Miscanthus* sp. are known to produce industrially important enzymes [58]. *Apiospora arundinis* and *Ap. saccharicola* isolated from a brown alga *Sargassum* sp. produce antimicrobial substances that can inhibit some plant pathogenic fungi [59]. The endophytic *Ap. rasikravindrae* was isolated from the stem of *Coleus amboinicus,* which produces a compound with strong antimicrobial and cytotoxic activities [60]. Eijk [61] reported that *Ap. sphaerosperma* produced a tetrahydroxy anthraquinone pigment and other metabolites, such as ergosterol, succinic acid, and phenolic compounds C18O5. Li et al. [62] conducted whole-genome sequencing of *Ap*. *sphaerosperma* and revealed the potential of *Ap*. *sphaerosperma* AP-Z13 to synthesize various secondary metabolites based on transcriptomics, proteomics, and metabolomics analyses. However, many novel *Apiospora* species, including new species in this study, are untapped natural resources and only *Ap*. *sphaerosperma* has been the subject of whole-gene sequencing and omics research [62]. The future necessitates further metabolomics analyses to investigate the biological applications of both known and newly discovered *Apiospora* species, in order to comprehensively explore their biological properties.

## Figures and Tables

**Figure 1 jof-09-01087-f001:**
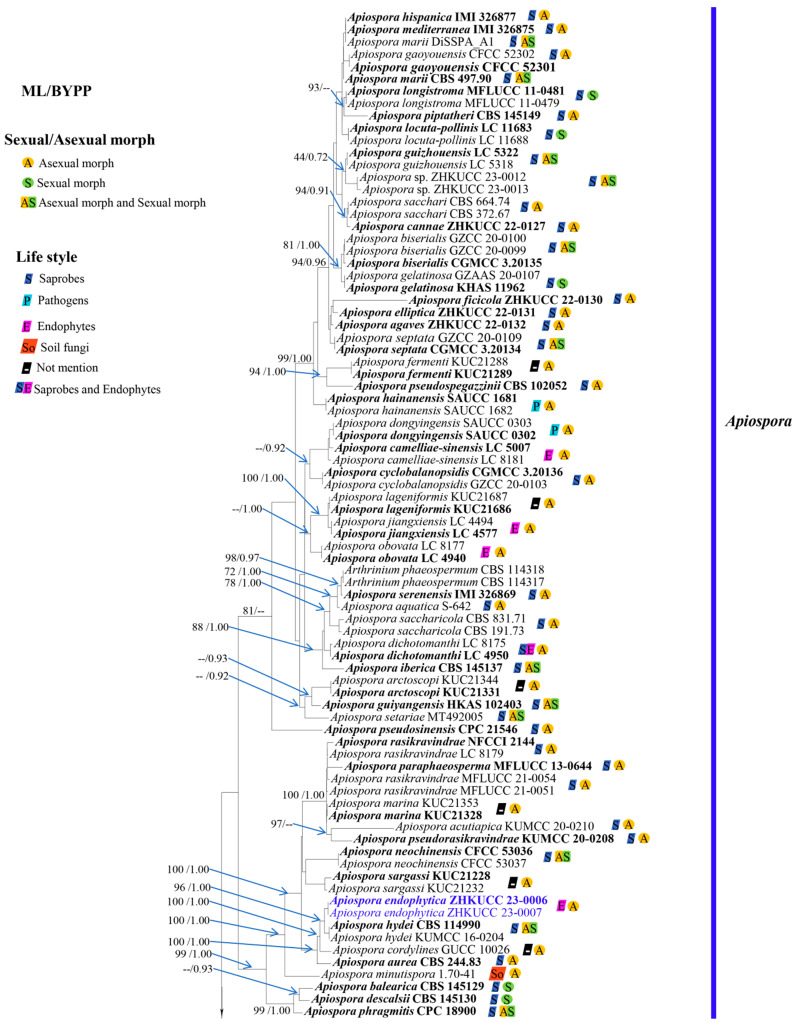
Phylogram generated from maximum likelihood analysis (RAxML) of genera in Apiosporaceae based on ITS, LSU, *tef1-α*, and *tub2* sequence data. Maximum likelihood bootstrap values equal or above 75%, and Bayesian posterior probabilities equal or above 0.90 (ML/BYPP) are given at the nodes. A strain number is noted after the species name. The tree is rooted with *Sporocadus trimorphus* (CFCC 55171) and (ROC 113). Hyphen (-) represents support values below 75% ML and 0.90 BYPP. The ex-type strains are bolded black, and the new isolates are in blue.

**Figure 2 jof-09-01087-f002:**
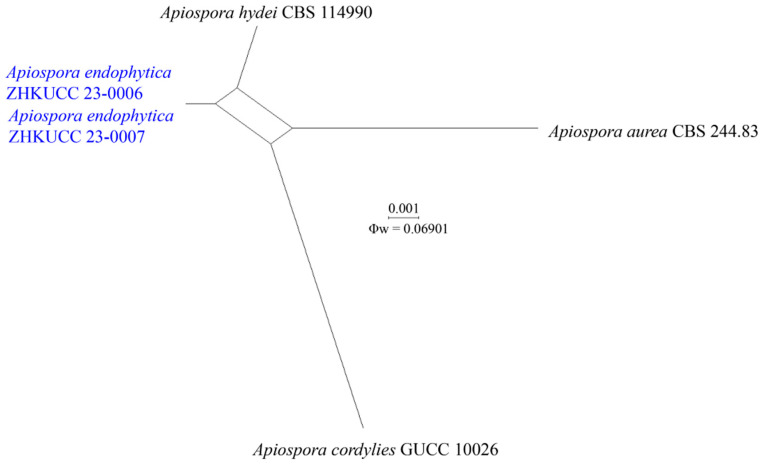
Split graph showing the results of the pairwise homoplasy index (PHI) test of the combined ITS, LSU, *tef1-α*, and *tub2* sequence data between *Apiospora endophytica* (ZHKU 23-0006, ZHKU 23-0007) with three closely related taxa of *A. aurea* CBS 244.83, *A*. *hydei* CBS 114990, and *A*. *cordylies* GUCC 10026 using LogDet transformation and splits decomposition. PHI test result (Φw) = 0.06901 indicates no significant recombination within the dataset (Φw > 0.05). The generated sequences are indicated in blue.

**Figure 3 jof-09-01087-f003:**
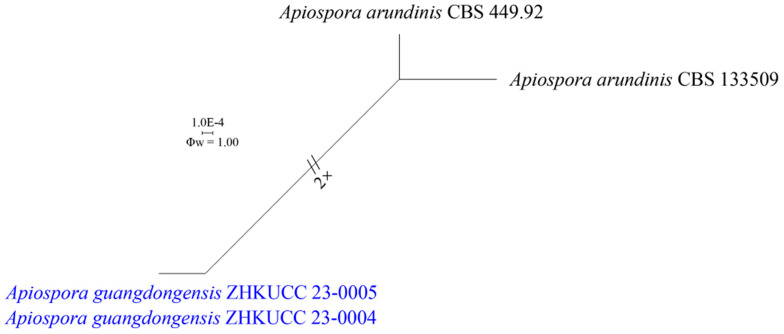
Split graph showing the results of the pairwise homoplasy index (PHI) test of the combined ITS, LSU, *tef1-α*, and *tub2* sequence data between *Apiospora guangdongensis* (ZHKUCC 23-0004, ZHKUCC 23-0005) with the closely related taxa of *A*. *arundinis* (CBS 449.92, CBS 133509) using LogDet transformation and splits decomposition. PHI test result (Φw) = 1.00 indicates no significant recombination within the dataset (Φw > 0.05). The generated sequences are indicated in blue.

**Figure 4 jof-09-01087-f004:**
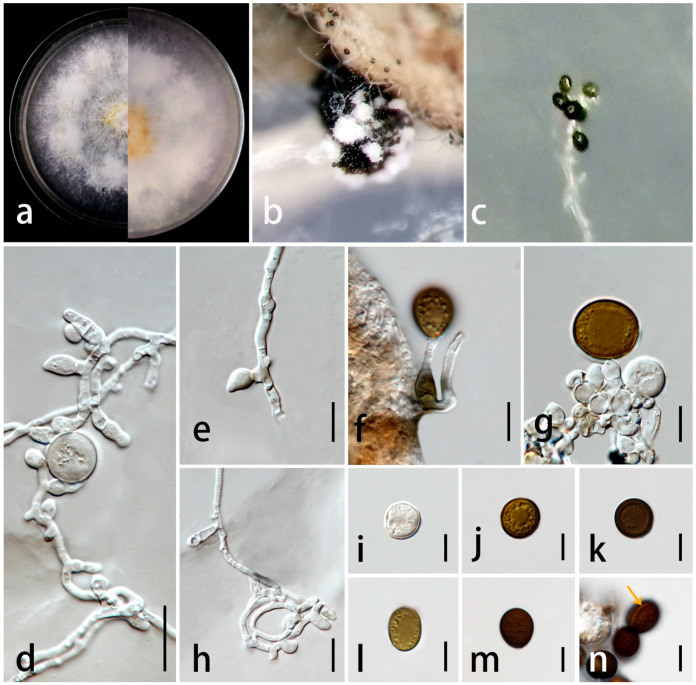
*Apiospora endophytica* (ZHKU 23-0002, holotype). (**a**) Upper view and reverse view of culture on PDA. (**b**,**c**) Conidia on aerial mycelia on PDA. (**d**–**h**) Conidiophores with conidiogenous cells. (**i**–**m**) Conidia in the face view. (**n**) Conidia with germ-slit. Scale bars in (**d**–**n**) = 10 μm.

**Figure 5 jof-09-01087-f005:**
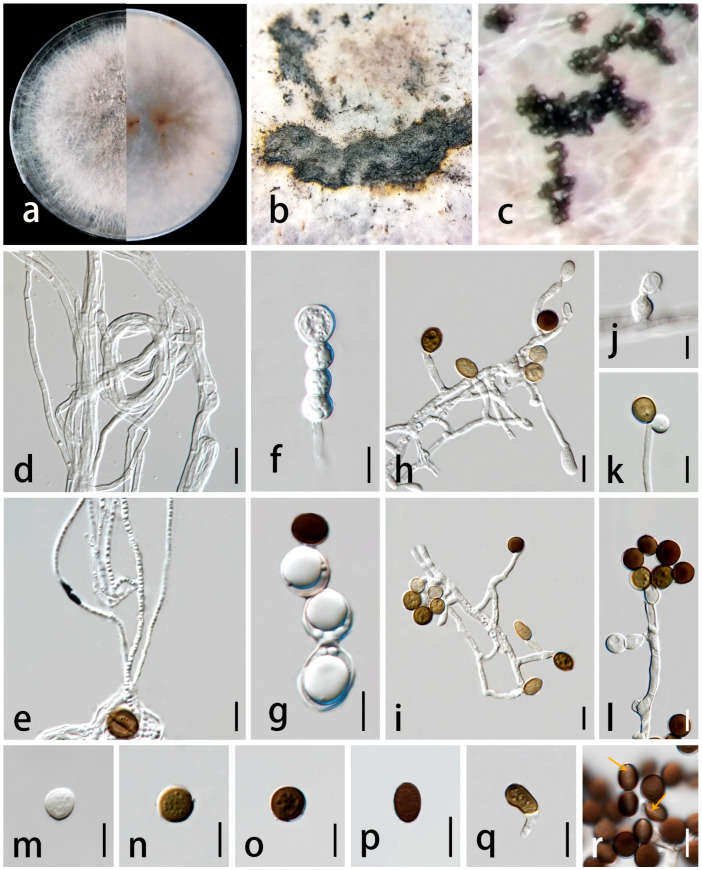
*Apiospora guangdongensis* (ZHKU 23-0001, holotype). (**a**) Upper view and reverse view of culture on PDA. (**b**,**c**) Conidia on aerial mycelia on PDA. (**d**,**e**) Mycelium. (**f**,**g**) Chlamydospores. (**h**–**l**) Conidiophores with conidiogenous cells. (**m**–**p**) Conidia in the face view. (**q**) Elongated conidia (sterile cells). (**r**) Conidia with germ-slit (arrows). Scale bars in (**d**–**r**) = 10 μm.

**Figure 6 jof-09-01087-f006:**
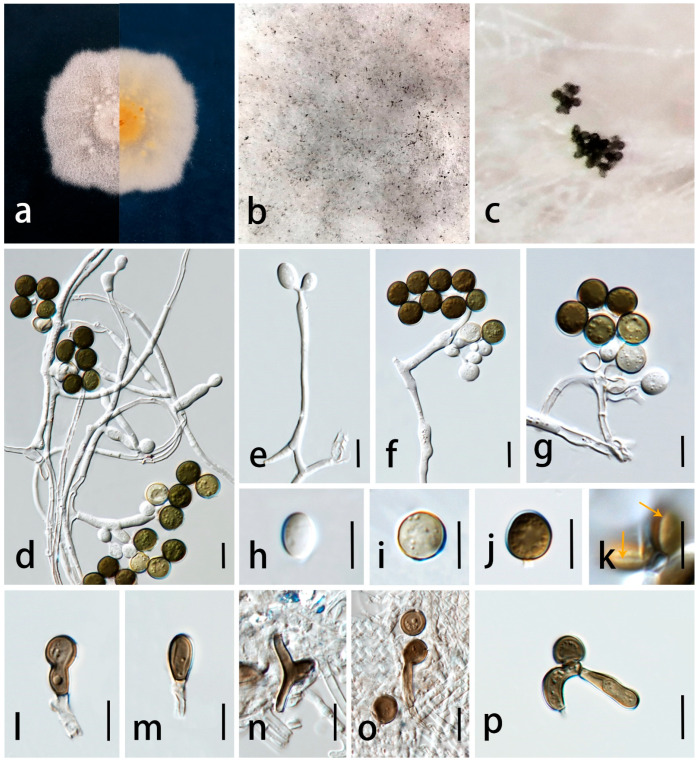
*Apiospora wurfbainiae* (ZHKU 23-0003, holotype). (**a**) Upper view and reverse view of culture on PDA. (**b**,**c**) Conidia on aerial mycelia on PDA. (**d**–**g**) Conidia with conidiogenous cells. (**d**–**j**) Conidia. (**k**) Conidia in the side view with germ-slit (arrows). (**l**–**n**) Sterile cells. (**o**,**p**) Sterile cell with conidia. Scale bars in (**d**–**p**) = 10 μm.

**Figure 7 jof-09-01087-f007:**
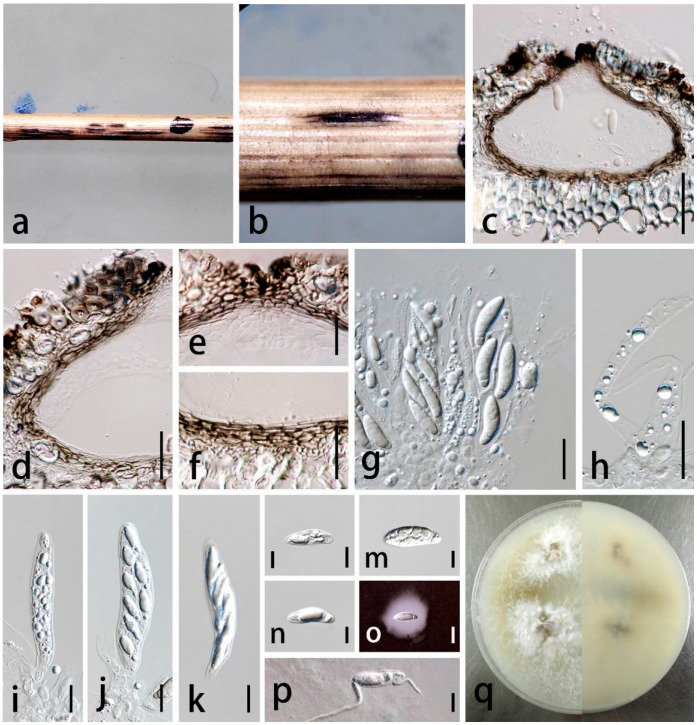
*Apiospora yunnanensis* (ZHKU 23-0004, holotype). (**a**,**b**) Appearance of ascomata on substrate. (**c**) Vertical section through ascoma. (**d**) Peridium. (**e**) Peridium at the top. (**f**) Peridium at the base. (**g**) Hamathecium with asci. (**h**) Hamathecium. (**i**–**k**) Asci. (**l**–**n**) Ascospores. (**o**) Ascospore in Indian Ink. (**p**) Germinated ascospore. (**q**) Culture characteristics on PDA (left-front, right-reverse). Scale bars in (**c**–**k**) = 20 μm, (**i**–**p**) = 10 μm.

**Table 1 jof-09-01087-t001:** Details of taxa including their GenBank accession numbers used in the phylogenetic analyses of this study.

Taxa	Strain Numbers	Substrates	Known Lifestyles	Countries	GenBank Accession Numbers			
					ITS	LSU	*tub2*	*tef1-α*
*Apiospora acutiapica*	KUMCC 20-0210	*Bambusa bambos*	Saprobe	China	MT946343	MT946339	MT947366	MT947360
*Ap. agari*	KUC21333^T^	*Agarum cribrosum*	Not mentioned	Republic of Korea	MH498520	-	MH498478	MH544663
*Ap. agari*	KUC21361	*Agarum cribrosum*	Not mentioned	Republic of Korea	MH498519	-	MH498477	MN868914
*Ap. aquatica*	S-642	Submerged wood	Saprobe	China	MK828608	MK835806	-	-
*Ap. arctoscopi*	KUC21331^T^	Egg of *Arctoscopus japonicus*	Not mentioned	Republic of Korea	MH498529	-	MH498487	MN868918
*Ap. arctoscopi*	KUC21344	Egg of *Arctoscopus japonicus*	Not mentioned	Republic of Korea	MH498528	-	MH498486	MN868919
*Ap. arundinis*	CBS 133509	*Aspergillus flavus* sclerotium	Saprobe/endophyte	USA	KF144886	KF144930	KF144976	KF145018
*Ap. arundinis*	CBS 449.92	*Aspergillus flavus* sclerotium	Saprobe/endophyte	USA	KF144887	KF144931	KF144977	KF145019
*Ap. aurea*	CBS 244.83^T^	-	Saprobe	Japan	AB220251	KF144935	KF144981	KF145023
*Ap. balearica*	CBS 145129^T^	Undetermined Poaceae	Saprobe	Spain	MK014869	MK014836	MK017975	MK017946
*Ap. bambusicola*	MFLUCC 20-0144^T^	*Schizostachyum brachycladum*	Saprobe	Thailand	MW173030	MW173087	-	MW183262
*Ap. biserialis*	CGMCC 3.20135^T^	Bamboo	Saprobe	China	MW481708	MW478885	MW522955	MW522938
*Ap. biserialis*	GZCC 20-0099	Bamboo	Saprobe	China	MW481709	MW478886	MW522956	MW522939
*Ap. biserialis*	GZCC 20-0100	Bamboo	Saprobe	China	MW481710	MW478887	MW522957	MW522940
*Ap. camelliae-sinensis*	LC 5007^T^	*Camellia sinensis*	Endophyte	China	KY494704	KY494780	KY705173	KY705103
*Ap. camelliae-sinensis*	LC 8181	*Camellia sinensis*	Endophyte	China	KY494761	KY494837	KY705229	KY705157
*Ap. chiangraiense*	MFLUCC 21-0053^T^	Dead culms of bamboo	Saprobe	Thailand	MZ542520	MZ542524	MZ546409	-
*Ap. chromolaenae*	MFLUCC 17-1505^T^	*Chromolaena odorata*	Saprobe	Thailand	MT214342	MT214436	-	MT235802
*Ap. cordylines*	GUCC 10026	*Cordyline fruticosa*	Not mentioned	China	MT040105	-	MT040147	MT040126
*Ap. cyclobalanopsidis*	CGMCC 3.20136^T^	*Cyclobalanopsidis glauca*	Saprobe	China	MW481713	MW478892	MW522962	MW522945
*Ap. cyclobalanopsidis*	GZCC 20-0103	*Cyclobalanopsidis glauca*	Saprobe	China	MW481714	MW478893	MW522963	MW522946
*Ap. descalsii*	CBS 145130^T^	*Ampelodesmos mauritanicus*	Saprobe	Spain	MK014870	MK014837	MK017976	MK017947
*Ap. dichotomanthi*	LC 4950^T^	*Dichotomanthes tristaniicarpa*	Saprobe/endophyte	China	KY494697	KY494773	KY705167	KY705096
*Ap. dichotomanthi*	LC 8175	*Dichotomanthes tristaniicarpa*	Saprobe/endophyte	China	KY494755	KY494831	KY705223	KY705151
*Ap. dongyingensis*	SAUCC 0302^T^	Leaf of bamboo	Pathogen	China	OP563375	OP572424	OP573270	OP573264
*Ap. dongyingensis*	SAUCC 0303	Leaf of bamboo	Pathogen	China	OP563374	OP572423	OP573263	OP573269
* Ap. endophytica *	ZHKUCC 23-0006^T^	* Wurfbainia villosa *	Endophyte	China	OQ587996	OQ587984	OQ586062	OQ586075
* Ap. endophytica *	ZHKUCC 23-0007	* Wurfbainia villosa *	Endophyte	China	OQ587997	OQ587985	OQ586063	OQ586076
*Ap. esporlensis*	CBS 145136^T^	*Phyllostachys aurea*	Saprobe	Spain	MK014878	MK014845	MK017983	MK017954
*Ap. euphorbiae*	IMI 285638b	*Bambusa* sp.	Saprobe	Bangladesh	AB220241	AB220335	AB220288	-
*Ap. fermenti*	KUC21289^T^	Seaweed	Not mentioned	Republic of Korea	MF615226	-	MF615231	MH544667
*Ap. fermenti*	KUC21288	Seaweed	Not mentioned	Republic of Korea	MF615230	-	MF615235	MH544668
*Ap. gaoyouensis*	CFCC 52301^T^	*Phragmites australis*	Saprobe	China	MH197124	-	MH236789	MH236793
*Ap. gaoyouensis*	CFCC 52302	*Phragmites australis*	Saprobe	China	MH197125	-	MH236790	MH236794
*Ap. garethjonesii*	KUMCC 16-0202^T^	Dead culms of bamboo	Saprobe	China	KY356086	KY356091	-	-
*Ap. gelatinosa*	KHAS 11962^T^	Bamboo	Saprobe	China	MW481706	MW478888	MW522958	MW522941
*Ap. gelatinosa*	GZAAS 20-0107	Bamboo	Saprobe	China	MW481707	MW478889	MW522959	MW522942
* Ap. guangdongensis *	ZHKUCC 23-0004^T^	* Wurfbainia villosa *	Endophyte	China	OQ587994	OQ587982	OQ586060	OQ586073
* Ap. guangdongensis *	ZHKUCC 23-0005	* Wurfbainia villosa *	Endophyte	China	OQ587995	OQ587983	OQ586061	OQ586074
*Ap. guiyangensis*	HKAS 102403^T^	Unidentified grass	Saprobe	China	MW240647	MW240577	MW775604	MW759535
*Ap. guizhouensis*	LC 5318	Air in karst cave, bamboo	Airborne/endophyte	China	KY494708	KY494784	KY705177	KY705107
*Ap. guizhouensis*	LC 5322^T^	Air in karst cave, bamboo	Airborne/endophyte	China	KY494709	KY494785	KY705178	KY705108
*Ap. hainanensis*	SAUCC 1681^T^	Leaf of bamboo	Pathogen	China	OP563373	OP572422	OP573268	OP573262
*Ap. hainanensis*	SAUCC 1682	Leaf of bamboo	Pathogen	China	OP563372	OP572421	OP573267	OP573261
*Ap. hispanica*	IMI 326877^T^	Beach sand	Saprobe	Spain	AB220242	AB220336	AB220289	-
*Ap. hydei*	CBS 114990^T^	Culms of *Bambusa tuldoides*	Saprobe	Hong Kong, China	KF144890	KF144936	KF144982	KF145024
*Ap. hydei*	KUMCC 16-0204	*Bambusa tuldoides*	Saprobe	China	KY356087	KY356092	-	-
*Ap. hyphopodii*	MFLUCC 15-0003^T^	*Bambusa tuldoides*	Saprobe	China	KR069110	-	-	-
*Ap. hyphopodii*	KUMCC 16-0201	*Bambusa tuldoides*	Saprobe	China	KY356088	KY356093	-	-
*Ap. hysterina*	ICPM 6889^T^	Bamboo	Saprobe	New Zealand	MK014874	MK014841	MK017980	MK017951
*Ap. hysterina*	CBS 145133	Bamboo	Saprobe	New Zealand	MK014875	MK014842	MK017981	MK017952
*Ap. iberica*	CBS 145137^T^	*Arundo donax*	Saprobe	Portugal	MK014879	MK014846	MK017984	MK017955
*Ap. intestini*	CBS 135835^T^	Gut of a grasshopper	Saprobe	India	KR011352	MH877577	KR011350	KR011351
*Ap. intestini*	MFLUCC 21-0052	Gut of a grasshopper	Saprobe	India	MZ542521	MZ542525	MZ546410	MZ546406
*Ap. italica*	CBS 145138^T^	*Arundo donax*	Saprobe	Italy	MK014880	MK014847	MK017985	MK017956
*Ap. italica*	CBS 145139	*Arundo donax*	Saprobe	Italy	MK014881	MK014848	MK017986	-
*Ap. jatrophae*	AMH-9557^T^	*Jatropha podagrica*	Saprobe	India	JQ246355	-	-	-
*Ap. jatrophae*	AMH-9556	*Jatropha podagrica*	Saprobe	India	HE981191	-	-	-
*Ap. jiangxiensis*	LC 4494	*Maesa* sp.	Endophyte	China	KY494690	KY494766	KY705160	KY705089
*Ap. jiangxiensis*	LC 4577^T^	*Maesa* sp.	Endophyte	China	KY494693	KY494769	KY705163	KY705092
*Ap. kogelbergensis*	CBS 113332	Dead culms of Restionaceae	Saprobe	South Africa	KF144891	KF144937	KF144983	KF145025
*Ap. kogelbergensis*	CBS 113333^T^	Dead culms of Restionaceae	Saprobe	South Africa	KF144892	KF144938	KF144984	KF145026
*Ap. koreana*	KUC21332^T^	Egg of *Arctoscopus japonicus*	Not mentioned	Republic of Korea	MH498524	-	MH498482	MH544664
*Ap. koreana*	KUC21348	Egg of *Arctoscopus japonicus*	Not mentioned	Republic of Korea	MH498523	-	MH498481	MN868927
*Ap. lageniformis*	KUC21686^T^	Branch of *Phyllostachys pubescens*	Not mentioned	Republic of Korea	ON764022	ON787761	ON806636	ON806626
*Ap. lageniformis*	KUC21687	Branch of *Phyllostachys pubescens*	Not mentioned	Republic of Korea	ON764023	ON787762	ON806637	ON806627
*Ap. locuta-pollinis*	LC 11688	*Brassica campestris*	Saprobe	China	MF939596	-	MF939623	MF939618
*Ap. locuta-pollinis*	LC 11683^T^	*Brassica campestris*	Saprobe	China	MF939595	-	MF939622	MF939616
*Ap. longistroma*	MFLUCC 11-0479	Dead culms of bamboo	Saprobe	Thailand	KU940142	KU863130	-	-
*Ap. longistroma*	MFLUCC 11-0481^T^	Dead culms of bamboo	Saprobe	Thailand	KU940141	KU863129	-	-
*Ap. magnispora*	ZHKUCC 22-0001	Bamboo	Saprobe	China	OM728647	OM486971	OM0543544	OM543543
*Ap. malaysiana*	CBS 102053^T^	*Macaranga hullettii*	Saprobe	Malaysia	KF144896	KF144942	KF144988	KF145030
*Ap. marianiae*	CBS 148710^T^	*Phleum pratense*	Saprobe	Spain	NR_183001	NG_149092		-
*Ap. marianiae*	AP301119	*Phleum pratense*	Saprobe	Spain	ON692407	ON692423	ON677187	ON677181
*Ap. marii*	CBS 497.90^T^	Beach sands	Saprobe	Spain	AB220252	KF144947	KF144993	KF145035
*Ap. marii*	DiSSPA_A1	Beach sands	Saprobe	Spain	MK602320	-	MK614695	MK645472
*Ap. marina*	KUC21328^T^	Seaweed	Not mentioned	Republic of Korea	MH498538	-	MH498496	MH544669
*Ap. marina*	KUC21353	Seaweed	Not mentioned	Republic of Korea	MH498537	-	MH498495	MN868923
*Ap. mediterranea*	IMI 326875^T^	Air	Saprobe	Spain	AB220243	AB220337	AB220290	-
*Ap. minutispora*	1.70-41	Mountain soil	Soil	Republic of Korea	LC517882	-	LC518888	LC518889
*Ap. mori*	MFLUCC 20-0181^T^	*Morus australis*	Saprobe	Taiwan	MW114313	MW114393	-	-
*Ap. mori*	NCYUCC 19-034	*Morus australis*	Saprobe	Taiwan	MW114314	MW114394	-	-
*Ap. mukdahanensis*	MFLUCC 22-0056^T^	dead bamboo leave	Saprobe	Thailand	OP377735	OP377742	-	OP381089
*Ap. multiloculata*	MFLUCC 21-0023^T^	Dead bamboo	Saprobe	Thailand	OL873137	OL873138	-	-
*Ap. mytilomorpha*	DAOM 214595^T^	*Andropogon* sp.	Saprobe	India	KY494685	-	-	-
*Ap. neobambusae*	LC 7106^T^	Leaves of bamboo	Saprobe/endophyte	China	KY494718	KY494794	KY705186	KY806204
*Ap. neobambusae*	LC 7124	Leaves of bamboo	Saprobe/endophyte	China	KY494727	KY494803	KY705195	KY806206
*Ap. neochinensis*	CFCC 53036^T^	*Fargesia qinlingensis*	Saprobe	China	MK819291	-	MK818547	MK818545
*Ap. neochinensis*	CFCC 53037	*Fargesia qinlingensis*	Saprobe	China	MK819292	-	MK818548	MK818546
*Ap. neogarethjonesii*	KUMCC 18-0192	Bamboo	Saprobe	China	MK070897	MK070898	-	-
*Ap. neosubglobosa*	JHB 006	Bamboo	Saprobe	China	KY356089	KY356094	-	-
*Ap. neosubglobosa*	KUMCC 16-0203^T^	Bamboo	Saprobe	China	KY356090	KY356095	-	-
*Ap. obovata*	LC 4940^T^	*Lithocarpus* sp.	Endophyte	China	KY494696	KY494772	KY705166	KY705095
*Ap. obovata*	LC 8177	*Lithocarpus* sp.	Endophyte	China	KY494757	KY494833	KY705225	KY705153
*Ap. ovata*	CBS 115042^T^	*Arundinaria hindsii*	Saprobe	China	KF144903	KF144950	KF144995	KF145037
*Ap. paraphaeosperma*	MFLUCC 13-0644^T^	Dead culms of bamboo	Saprobe	Thailand	KX822128	KX822124	-	-
*Ap. phragmitis*	CPC 18900^T^	*Phragmites australis*	Saprobe	Italy	KF144909	KF144956	KF145001	KF145043
*Ap. phyllostachydis*	MFLUCC 18-1101^T^	*Phyllostachys heteroclada*	Saprobe	China	MK351842	MH368077	MK291949	MK340918
*Ap. piptatheri*	CBS 145149^T^	*Piptatherum miliaceum*	Saprobe	Spain	MK014893	MK014860	-	MK017969
*Ap. pseudohyphopodii*	KUC21680^T^	Culm of *Phyllostachys pubescens*	Not mentioned	Republic of Korea	ON764026	ON787765	ON806640	ON806630
*Ap. pseudohyphopodii*	KUC21684	Culm of *Phyllostachys pubescens*	Not mentioned	Republic of Korea	ON764027	ON787766	ON806641	ON806631
*Ap. pseudoparenchymatica*	LC 7234^T^	Leaves of bamboo	Endophyte	China	KY494743	KY494819	KY705211	KY705139
*Ap. pseudoparenchymatica*	LC 8173	Leaves of bamboo	Endophyte	China	KY494753	KY494829	KY705221	KY705149
*Ap. pseudorasikravindrae*	KUMCC 20-0208^T^	*Bambusa dolichoclada*	Saprobe	China	MT946344	-	MT947367	MT947361
*Ap. pseudosinensis*	CPC 21546^T^	Leaves of bamboo	Saprobe	Netherlands	KF144910	KF144957	-	KF145044
*Ap. pseudospegazzinii*	CBS 102052^T^	*Macaranga hullettii*	Saprobe	Malaysia	KF144911	KF144958	KF145002	KF145045
*Ap. pterosperma*	CBS 123185	*Lepidosperma gladiatum*	Saprobe	Australia	KF144912	KF144959	KF145003	-
*Ap. pterosperma*	CPC 20193^T^	*Lepidosperma gladiatum*	Saprobe	Australia	KF144913	KF144960	KF145004	KF145046
*Ap. pusillisperma*	KUC21321^T^	Seaweed	Not mentioned	Republic of Korea	MH498533	-	MH498491	MN868930
*Ap. pusillisperma*	KUC21357	Seaweed	Not mentioned	Republic of Korea	MH498532	-	MH498490	MN868931
*Ap. qinlingensis*	CFCC 52303^T^	*Fargesia qinlingensis*	Saprobe	China	MH197120	-	MH236791	MH236795
*Ap. qinlingensis*	CFCC 52304	*Fargesia qinlingensis*	Saprobe	China	MH197121	-	MH236792	MH236796
*Ap. rasikravindrae*	LC 8179	*Brassica rapa*	Saprobe	China	KY494759	KY494835	KY705227	KY705155
*Ap. rasikravindrae*	NFCCI 2144^T^	Soil	Saprobe	Norway	JF326454	-	-	-
*Ap. rasikravindrae*	MFLUCC 21-0051	Dead culms of bamboo	Saprobe	Thailand	MZ542523	MZ542527	MZ546412	MZ546408
*Ap. rasikravindrae*	MFLUCC 21-0054	Dead culms of Maize	Saprobe	Thailand	MZ542522	MZ542526	MZ546411	MZ546407
*Ap. sacchari*	CBS 372.67	Air	Endophyte	-	KF144918	KF144964	KF145007	KF145049
*Ap. sacchari*	CBS 664.74	Soil under *Calluna vulgaris*	Endophyte	Netherlands	KF144919	KF144965	KF145008	KF145050
*Ap. saccharicola*	CBS 191.73	Air	Endophyte	Netherlands	KF144920	KF144966	KF145009	KF145051
*Ap. saccharicola*	CBS 831.71	-	Endophyte	Netherlands	KF144922	KF144969	KF145012	KF145054
*Ap. sargassi*	KUC21228^T^	*Sargassum fulvellum*	Not mentioned	Republic of Korea	KT207746	-	KT207644	MH544677
*Ap. sargassi*	KUC21232	*Sargassum fulvellum*	Not mentioned	Republic of Korea	KT207750	-	KT207648	MH544676
*Ap. sasae*	CBS 146808^T^	dead culms	Saprobe	Netherlands	MW883402	MW883797	MW890120	MW890104
*Ap. septata*	CGMCC 3.20134^T^	bamboo	Saprobe	China	MW481711	MW478890	MW522960	MW522943
*Ap. septata*	GZCC 20-0109	bamboo	Saprobe	China	MW481712	MW478891	MW522961	MW522944
*Ap. serenensis*	IMI 326869^T^	excipients, atmosphere andhome dust	Saprobe	Spain	AB220250	AB220344	AB220297	-
*Ap. setariae*	MT492005	*Setaria viridis*	Saprobe	China	MT492005	-	MT497467	MW118457
*Ap. setostroma*	KUMCC 19-0217^T^	Dead branches of bamboo	Saprobe	China	MN528012	MN528011	-	MN527357
*Ap. sichuanensis*	HKAS 107008^T^	dead culm of grass	Saprobe	China	MW240648	MW240578	MW775605	MW759536
*Ap. sorghi*	URM 93000^T^	*Sorghum bicolor*	Endophyte	Brazil	MK371706	-	MK348526	-
*Ap.* sp.	ZHKUCC 23-0010	*Wurfbainia villosa*	Endophyte	China	OQ588000	OQ587988	OQ586066	OQ586079
*Ap.* sp.	ZHKUCC 23-0011	*Wurfbainia villosa*	Endophyte	China	OQ588001	OQ587989	OQ586067	OQ586080
*Ap.* sp.	ZHKUCC 23-0012	*Wurfbainia villosa*	Endophyte	China	OQ588002	OQ587990	OQ586068	OQ586081
*Ap.* sp.	ZHKUCC 23-0013	*Wurfbainia villosa*	Endophyte	China	OQ588003	OQ587991	OQ586069	OQ586082
*Ap. stipae*	CBS 146804^T^	dead culm of *Stipa gigantea*	Saprobe	Spain	MW883403	MW883798	MW890121	MW890082
*Ap. subglobosa*	MFLUCC 11-0397^T^	Dead culms of bamboo	Saprobe	Thailand	KR069112	KR069113	-	-
*Ap. subrosea*	LC 7291	Leaves of bamboo	Endophyte	China	KY494751	KY494827	KY705219	KY705147
*Ap. subrosea*	LC 7292^T^	Leaves of bamboo	Endophyte	China	KY494752	KY494828	KY705220	KY705148
*Ap. taeanensis*	KUC21322^T^	Seaweed	Not mentioned	Republic of Korea	MH498515	-	MH498473	MH544662
*Ap. taeanensis*	KUC21359	Seaweed	Not mentioned	Republic of Korea	MH498513	-	MH498471	MN868935
*Ap. thailandica*	MFLUCC 15-0199	Dead culms of bamboo	Saprobe	Thailand	KU940146	KU863134	-	-
*Ap. thailandica*	MFLUCC 15-0202^T^	Dead culms of bamboo	Saprobe	Thailand	KU940145	KU863133	-	-
*Ap. tropica*	MFLUCC 21-0056^T^	Dead culms of bamboo	Saprobe	Thailand	OK491657	OK491653	OK560922	-
*Ap. vietnamensis*	IMI 99670^T^	*Citrus sinensis*	Saprobe	Vietnam	KX986096	KX986111	KY019466	-
* Ap. wurfbainiae *	ZHKUCC 23-0008^T^	* Wurfbainia villosa *	Endophyte	China	OQ587998	OQ587986	OQ586064	OQ586077
* Ap. wurfbainiae *	ZHKUCC 23-0009	* Wurfbainia villosa *	Endophyte	China	OQ587999	OQ587987	OQ586065	OQ586078
*Ap. xenocordella*	CBS 478.86^T^	Soil from roadway	Soil	Zimbabwe	KF144925	KF144970	KF145013	KF145055
*Ap. xenocordella*	CBS 595.66	On dead branches	Saprobe	Misiones	KF144926	KF144971	-	-
*Ap. yunnana*	DDQ 00281	*Phyllostachys nigra*	Saprobe	China	KU940148	KU863136	-	-
*Ap. yunnana*	MFLUCC 15-1002^T^	*Phyllostachys nigra*	Saprobe	China	KU940147	KU863135	-	-
* Ap. yunnanensis *	ZHKUCC 23-0014^T^	Grass	Saprobe	China	OQ588004	OQ587992	OQ586070	OQ586083
* Ap. yunnanensis *	ZHKUCC 23-0015	Grass	Saprobe	China	OQ588005	OQ587993	OQ586071	OQ586084
*Arthrinium austriacum*	GZU 345004	*Carex pendula*	Saprobe	Austria	MW208928	-	-	-
*Ar. austriacum*	GZU 345006	*Carex pendula*	Saprobe	Austria	MW208929	MW208860	-	-
*Ar. sporophleum*	GZU 345102	*Carex firma*	Saprobe	Austria	MW208944	MW208866	-	-
*Ar. caricicola*	CBS 145127	*Carex ericetorum*	Saprobe	China	MK014871	MK014838	MK017977	MK017948
*Ar. crenatum*	AG19066^T^	Dead leaves of grass (probably *Festuca burgundiana*)	Saprobe	France	MW208931	MW208861	-	-
*Ar. curvatum*	AP 25418	Leaves of *Carex* sp.	Saprobe	China	MK014872	MK014839	MK017978	MK017949
*Ar. japonicum*	IFO 30500	-	Saprobe	Japan	AB220262	AB220356	AB220309	-
*Ar. japonicum*	IFO 31098	Leaves of *Carex despalata*	Saprobe	Japan	AB220264	AB220358	AB220311	-
*Ar. luzulae*	AP7619-3^T^	*Luzula sylvatica*	Saprobe	Spain	MW208937	MW208863	-	-
*Ar. morthieri*	GZU 345043	*Carex pilosa*	Saprobe	Austria	MW208938	MW208864	-	-
*Ar. phaeospermum*	CBS 114317	Leaves of *Hordeum vulgare*	Saprobe	Iran	KF144906	KF144953	KF144998	KF145040
*Ar. phaeospermum*	CBS 114318	Leaves of *Hordeum vulgare*	Saprobe	Iran	KF144907	KF144954	KF144999	KF145041
*Ar. puccinioides*	CBS 549.86	*Lepidosperma gladiatum*	Saprobe	Germany	AB220253	AB220347	AB220300	-
*Ar. sphaerospermum*	CBS 146355	Probably on Poaceae	Saprobe	Norway	MW208943	MW208865	-	-
*Ar. sporophleum*	CBS 145154	Dead leaves of *Juncus* sp.	Saprobe	Spain	MK014898	MK014865	MK018001	MK017973
*Ar. trachycarpum*	CFCC 53039	*Trachycarpus fortune*	Pathogen	China	MK301099	-	MK303395	MK303397
*Ar. urticae*	IMI 326344	*-*	Saprobe	-	AB220245	AB220339	AB220292	-
*Nigrospora aurantiaca*	CGMCC 3.18130^T^	*Nelumbo* sp.	Saprobe	China	KX986064	KX986098	KY019465	KY019295
*N. camelliae-sinensis*	CGMCC 3.18125^T^	*Camellia sinensis*	Endophyte/pathogen	China	KX985986	KX986103	KY019460	KY019293
*N. chinensis*	CGMCC 3.18127^T^	*Machilus breviflora*	Endophyte/pathogen	China	KX986023	KX986107	KY019462	KY019422
*N. gorlenkoana*	CBS 480.73	*Vitis vinifera*	Endophyte/pathogen	Kazakhstan	KX986048	KX986109	KY019456	KY019420
*N. guilinensis*	CGMCC 3.18124^T^	*Camellia sinensis*	Endophyte/pathogen	China	KX985983	KX986113	KY019459	KY019292
*N. hainanensis*	CGMCC 3.18129^T^	*Musa paradisiaca*	Endophyte/pathogen	China	KX986091	KX986112	KY019464	KY019415
*N. lacticolonia*	CGMCC 3.18123^T^	*Camellia sinensis*	Endophyte/pathogen	China	KX985978	KX986105	KY019458	KY019291
*N. musae*	CBS 319.34	*Musa* sp.	Endophyte/pathogen	Australia	MH855545	KX986110	KY019455	KY019419
*N. oryzae*	LC2693	*Neolitsea* sp.	Saprobe	China	KX985944	KX986101	KY019471	KY019299
*N. osmanthi*	CGMCC 3.18126^T^	*Hedera nepalensis*	Endophyte/pathogen	China	KX986010	KX986106	KY019461	KY019421
*N. pyriformis*	CGMCC 3.18122^T^	*Citrus sinensis*	Endophyte/pathogen	China	KX985940	KX986100	KY019457	KY019290
*N. rubi*	LC2698^T^	*Rubus* sp.	Endophyte/pathogen	China	KX985948	KX986102	KY019475	KY019302
*N. sphaerica*	LC7298	*Nelumbo* sp.	Saprobe	China	KX985937	KX986097	KY019606	KY019401
*N. vesicularis*	CGMCC 3.18128^T^	*Musa paradisiaca*	Endophyte	China	KX986088	KX986099	KY019463	KY019294
*Sporocadus trimorphus*	CFCC 55171^T^	Rose	Not mentioned	China	OK655798	OK560389	OM401677	OL814555
*S. trimorphus*	ROC 113	Rose	Not mentioned	China	OK655799	OK560390	OM401678	OL814556

Notes: Newly generated sequences in this study are in blue. “T” indicates ex-type. “-” = information not available. Abbreviations: AMH: Ajrekar Mycological Herbarium, Pune, Maharashtra, India; AP: Alvarado Pintos; CBS: Westerdijk Fungal Biodiversity Institute, Utrecht, Netherlands; CFCC: China Forestry Culture Collection Center, Beijing, China; CGMCC: China General Micro biological Culture Collection; CPC: Culture collection of Pedro Crous, housed at the Westerdijk Fungal Biodiversity Institute; DAOM: Canadian Collection of Fungal Cultures, Ottawa, Canada; GUCC: Guizhou University Culture Collection, Guizhou, China; GZAAS: Guizhou Academy of Agricultural Sciences herbarium, China; GZCC: Guizhou Culture Collection, China; GZU: University of Graz, Austria; HKAS: Herbarium of Cryptogams, Kunming Institute of Botany, Chinese Academy of Sciences, Yunnan, China; ICMP: International Collection of Microorganisms from Plants, New Zealand; IFO: Institute for Fermentation, Osaka, Japan; IMI: Culture collection of CABI Europe UK Centre, Egham, UK; JHB: H.B. Jiang; KUC: the Korea University Fungus Collection, Seoul, Korea; SFC the Seoul National University Fungus Collection; KUMCC: Culture collection of Kunming Institute of Botany, Yunnan, China; LC: Personal culture collection of Lei Cai, housed in the Institute of Microbiology, Chinese Academy of Sciences, China; MFLUCC: Mae Fah Luang University Culture Collection, Chiang Rai, Thailand; NFCCI: National Fungal Culture Collection of India; SAUCC: Shandong Agricultural University Culture Collection.

**Table 2 jof-09-01087-t002:** Synopsis of morphological characteristics of *Ap. endophytica* and its closely related species.

Characters	*Apiospora* Species			
	*Ap*. *endophytica*	*Ap*. *hydei*	*Ap. cordylines*	*Ap. aurea*
Host/substrate	Asymptomatic leaf of *Wurfbainia villosa*	Culms of *Bambusa tuldoides*	Leaves of *Cordyline fruticosa*	Air
Conidiophores	Reduced to conidiogenous cells	Pale brown, smooth, subcylindrical, transversely septate, branched, 20–40 × 3–5 μm	NA	NA
Conidiogenous cells	Aggregated in clusters or solitary, hyaline to golden brown, smoothly, erect, unbranched, cylindrical or clavate, ampulliform or obtriangular, 4–14 × 2–7 μm ($\bar{X}$ = 7.5 × 5 μm)	Aggregated in clusters, brown, smooth, subcylindrical to doliiform to lageniform, 5–8 × 4–5 μm	Erect, aggregated into clusters, hyaline to pale brown, smooth, doliiform to ampulliform or lageniform, (3–)5–10(–15) × 2.6–5.3 µm ($\bar{X}$ = 7.0 × 4.5 µm)	Integrated, polyblastic, denticulate
Conidia	Initially hyaline, becoming pale brown to dark brown, globose to subglobose, obovoid to ellipsoidal in the face view, lenticular with a thick equatorial slit in the side view, smooth-walled, 14–19 × 12–18 μm ($\bar{X}$ = 17 × 15 μm, n = 30) in the face view, 11–19 × 9–16 μm ($\bar{X}$ = 15 × 12 μm, n = 20)	Brown, roughened, globose in face view, lenticular in the side view, with pale equatorial slit, (15–)17– 19(–22) μm diam. in face view, (10–)11–12(–14) μm diam. in the side view, with a central scar, 1.5–2 μm diam.	Olivaceous to brown, smooth to finely roughened, subglobose to ellipsoidal, 15–19 × 12.5–18.5 µm ($\bar{X}$ = 17.5 × 15.7 µm)	Solitary, terminal, and sometimes also lateral with a hyaline rim, brown or dark brown, smooth, aseptate, 10–30 × 10–15 μm
Reference	This study	[8]	[49]	[50]

NA: undetermined.

## Data Availability

Not applicable.

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
