# Peer review of "Taxonomic and Phylogenetic Updates on Apiospora: Introducing Four New Species from Wurfbainia villosa and Grasses in China"

_jof, 2023, doi:10.3390/jof9111087_

Round 1

Reviewer 1 Report

Comments and Suggestions for Authors

The authors are proposing four new species in the genus Apiospora isolated from plant tissues in China.  The paper is well written and organized, and phylogenetic analyses and figures are excellent.  Only minor comments.  

Page 1 Line 18 change AND HAVE MOSTLY BEEN RPORTED to “reported in”

Page 1 Line 23 change INTRODUCED to “proposed”

Page 1 Line 24 delete THEY ARE

Page 1 Line 27 change 4 NEW SPECIES to “Asia”

Page 1 Line 31 delete NUMEROUS FUNGAL GENERAL, E.G.,

Page 1 Line 58 – 63 give scientific and common name for all the host plants e.g., peach (Prunus persica)…as you did in line 68

Page 1 Line 65 do not capitalize Onychomycosis.

Comments on the Quality of English Language

n/a

Author Response

Dear editor and reviewer(s),

Thank you very much for your review regarding our manuscript entitled “Updating the genus Apiospora by introducing four new species isolated from Wurfbainia villosa and grasses in China”. We sincerely appreciate the valuable comments and corrections to improve the quality of our research. The manuscript has been revised and modified in accordance with the referees' critiques. The revised texts have been visually emphasized in yellow. We hope that the revised version of our manuscript meets the requirements for publication in the journal.

Reviewer 2 Report

Comments and Suggestions for Authors

The authors introduce four new species in Apiospora based on morphological and molecular data. The novelty of the strains is adequately supported. However, the manuscript has some issues that need to be addressed.

Some major comments: Discussion needs to be revised in order to present the additional findings produced by the authors and how they contribute in the taxonomic status of Apiospora. Therefore, the title should also be reconsidered. The manuscript has many tables. Some should be presented as supplementary material.

Detailed comments can be found in the manuscript.

Comments on the Quality of English Language

The language needs checking. Several sentences need to be corrected in terms of grammar and syntax. Some they have been highlighted but the entire manuscripts needs careful revision.

Author Response

(The authors gave the same response as above.)

Reviewer 3 Report

Comments and Suggestions for Authors

The research in itself - is well planned and performed.  The results are sound. As such it is worthy of publication.

However the presentation calls for some rewriting:

1.      Tables 3 and 4 should be presented, along with their reasoning, in the Results chapter. I suggest following Figure 1 as the data emanates from analyzing that figure. Only the consequences should be summed up in the Discussion.

2.     The declared conidial slits are not very clear. I would suggest adding an arrow pointing to a slit in Figs. 4n and 5r.

3.     The English is not always clear and accurate.  

4.     Two additional references numbered 1 and 2 exist, in the reference list, between Ref 31 and 32.  Where are these cited?

Comments on the Quality of English Language

 The English is not always clear and accurate.  I started pointing to such instances in the text, but gave up. 
I suggest a thorough English editing by an expert in fungal taxonomy.

Author Response

(The authors gave the same response as above.)
